# Uncertainty Aware Deployment of Pre-trained Task Conditioned Imitation Learning Policies

**Bo Wu, Bruce D. Lee, Bernadette Bucher, Nikolai Matni**

University of Pennsylvania

{*bobwu, brucele, bucherb, nmatni*}@seas.upenn.edu

**Abstract:** Large scale robotic policies that are trained to perform diverse tasks on many robotic platforms hold great promise; however, reliable generalization remains a major challenge. To address this challenge, it is crucial to appropriately calibrate these models and use the calibrated model to make uncertainty aware decisions across large and diverse sets of data. In this work, we propose an approach to achieve this in pre-trained language conditioned imitation learning agents. Our simulation results using the proposed approach on the pre-trained Perceiver-Actor model demonstrate its effectiveness at improving task completion rates. The code is available at: https://github.com/BobWu1998/uncertainty_quant_peract.git

**Keywords:** Imitation Learning, Uncertainty Quantification, Generalist Robotics

## 1 Introduction

Inspired by general purpose natural language processing and computer vision models, the robotics community has dedicated significant effort to train generalist decision making agents that operate across various robotic platforms and perform a wide array of tasks [1]. These so-called "foundation models" are typically trained once on a diverse dataset, then deployed in the desired setting with minimal task specific fine-tuning. These pre-trained models shown the ability to generalize in many ways such as to new tasks, new robot platforms, and new environments [2, 3, 4, 5].

However, the limits of the generalization ability for these decision making agents are not well understood. A primary challenge hindering this understanding of generalization limits and associated performance expectations is the lack of well-calibrated, task-specific uncertainty estimates, which precludes the possibility of uncertainty aware decision-making. In this work, we propose combining classification calibration techniques with an uncertainty aware decision making protocol to boost the success rate for task conditioned imitation learning (IL) policies. Moreover, the calibration results alone enhance understanding of the expected generalization performance of pre-trained models.

### 1.1 Related Work

**Generalist Robotic Policies**   Early efforts by Dasari et al. [2] introduced a diverse and extensive robotics dataset, and use this dataset to train visual foresight and supervised inverse models. Subsequent work uses such datasets to enable sample efficient IL [3]. Recent developments have embraced transformer-based architecture to train multi-modal decision making agents [6] and language conditioned IL policies for robotic manipulation [4, 5, 7]. Efforts have also been made to extend to more general prompt conditioning using images and videos [8]. Shah et al. [9] explored the development of a general visual navigation model. For a review of generalist decision making agents, see Yang et al. [1]. Our work improves the performance of these generalist robotic policies by incorporating calibrated uncertainty estimates to enable more informed decisions at inference time.

**Uncertainty Quantification** For robotic decision making, learned models are often desired to provide a confidence score quantifying the model's uncertainty in its output, enabling the downstream formulation of safe decisions in robotic planning problems, as in e.g. [10, 11]. However, it has been shown that the confidence scores output by modern classification models are poorly calibrated in that they are not representative of the true correctness likelihood [12]. While numerous approaches to improving calibration post-hoc exist, such as temperature scaling [12] and conformal prediction [13], these techniques are seldom applied in the realm of pre-trained task-conditioned imitation learning policies to enhance generalization performance. A notable exception is the recent work by Ren et al. [14] which applies conformal prediction to language-based planners to request operator clarification in the face of ambiguity. In contrast, we consider a setting in which there is no ambiguity in the task conditioning, but rather the pre-trained model is imperfect. We calibrate the uncertainty of this pre-trained model, and use it to select actions in an uncertainty aware manner.

## 1.2 Contributions

We propose a straightforward yet effective modification to enhance the generalization[1] capabilities of pre-trained task-conditioned imitation learning agents, comprised of two components:

- A calibration step, tailored to the task of interest, that refines the model's outputs to generate confidence scores that approximate the correctness likelihood for imitating the expert.
- An uncertainty aware action selection technique using the probability distribution output by the calibrated model (See Algorithm 1).

By instantiating our approach on the PerAct model trained by Shridhar et al. [7] *without additional training*, we show that using a small expert calibration dataset from the target task enables a 3% improvement in task completion rate, particularly benefiting tasks with the lowest completion rate.

## 2 Approach

### 2.1 Background: Task-Conditioned Imitation Learning for Robotic Manipulation

We consider robotic manipulation problems in a discrete time decision making context. Let $o_t \in \mathcal{O}$ be observation of the world at time $t$ and $a_t \in \mathcal{A}$ be an action applied to the robot at time $t$, with $\mathcal{A}$ being a finite, discretized action space. We assume access to $n$ expert demonstrations $\{\zeta_1, \ldots, \zeta_n\}$ accompanied by task conditioning $\{c_1, \ldots, c_n\}$. Each demonstration consists of a trajectory of observations, and the corresponding action taken by an expert. The task conditioning $c_i$ provides information about the task which the corresponding expert demonstration is completing. The goal in task conditioned imitation learning is to find a policy which imitates the expert behavior for any given task. As the action space is discrete, the policy operates as a classifier for the observation-task conditioning pair. In particular, given some observation $o \in \mathcal{O}$ and task conditioning $c$, the policy selects an action as $\hat{\pi}(o, c) = \text{argmax}_{a' \in \mathcal{A}} \hat{f}_{j(a')}(o, c)$. Here, $j$ is a function $j : \mathcal{A} \to [|\mathcal{A}|]$ enumerating $\mathcal{A}$, and $\hat{f}(o, c)$ is a vector of logits with $\hat{f}_{j(a')}(o, c)$ representing its $j(a')^{\text{th}}$ element. Such a policy may be found through behavior cloning by choosing $\hat{f}$ belonging to some function class to minimize the cross entropy loss. Once the model is trained, it is deployed to select the robot's actions in the face of a target task $c_{n+1}$ as $a_t = \hat{\pi}(o_t, c_{n+1})$. In general, the target task may not be in the training set; however, our experiments consider the setting where $c_{n+1} \in \{c_1, \ldots, c_n\}$.

### 2.2 Target Task Calibration

The logits may be used to represent a probability distribution $p$ over possible actions by applying the softmax function: $p = \sigma_{SM}(\hat{f}(o, c))$. While $p$ is a viable probability distribution, it may not be well-aligned with the true correctness likelihood, especially for the the specific task $c_{n+1}$.

---

[1]Generalization has multiple meanings in the context of multi-task robotics. In this work, we focus on generalizing the pre-trained model to the same tasks in the training set but with different environment initializations.

To improve the alignment between the distribution $p$ and the correctness likelihood for task $c_{n+1}$, we collect a calibration set from task in question. The set consists of several demonstrations $\{\zeta_{n+1}, \ldots, \zeta_{n+k}\}$ along with their associated conditonings $\{c_{n+1}, \ldots, c_{n+k}\}$, where $c_{n+1} = \cdots = c_{n+k}$. Calibrated models may be obtained from this set with various methods, but we propose using temperature scaling [12]. Specifically, we determine the optimal temperature parameter $\hat{T}$ as the value of $T$ which minimizes the cross entropy loss of the model $\frac{1}{T}\hat{f}$ over the calibration set. Given this temperature parameter, we define the calibrated model outputting the probability scores as $p = \tilde{f}(o, c) = \sigma_{SM}(\hat{f}(o, c)/\hat{T})$ with components $p_k = \tilde{f}_k(o, c)$.

## 2.3 Uncertainty Aware Action Selection

The greedy approach outlined in Section 2.1 does not make use of the calibrated confidence scores from Section 2.2, and overlooks important structure of the action space. Specifically, distinct elements in $\mathcal{A}$ may represent similar actions. This structure should be leveraged in the manipulation problem, as the success of specific manipulation tasks are often robust to small deviations in these action spaces, e.g. a robotic gripper can open a drawer by pulling at different positions of the handle.

We propose Algorithm 1 to remedy the above shortcoming. Rather than selecting the action with the highest confidence score, Algorithm 1 selects the action for which the neighboring region of potential actions has the highest sum of confidence scores. The neighboring region of an action $a \in \mathcal{A}$ is $\left\{ a' : \|a - a'\|^2 \leq \tau \right\}$, where the norm is problem specific to capture similarity between two actions in $\mathcal{A}$, and $\tau$ is a robustness threshold. Note that while the greedy action selection approach is only sensitive to the ordering of highest confidence score output by the model, Algorithm 1 requires a well-calibrated model to adequately determine the confidence of the neighboring region of actions.

---

**Algorithm 1** Uncertainty Aware Action Selection

---

**Input:** Calibrated model $\tilde{f}$, conditioning $c$, observation $o$, robustness threshold $\tau$
Compute $p_1, \ldots, p_{|\mathcal{A}|} = \tilde{f}(o; c)$.        $\triangleright$ Retrieve confidence scores for actions
Set $\hat{a} = \text{argmax}_{a \in \mathcal{A}} \sum_{a' \in \mathcal{A}} p_{j(a')}\mathbf{1}(\|a - a'\|^2 < \tau)$    $\triangleright$ Action maximizing nearby confidence
**Output:** Uncertainty-aware action $\hat{a}$

---

# 3 Experimental Results

We consider the PerAct model from Shridhar et al. [7]. The action space consists of three coordinates for the discretized desired end effector translation, three coordinates for the discretized desired end effector rotation, and two binary values indicating gripper status and collision avoidance module activation. The desired action is then achieved using a sampling-based motion planner. The world state is represented by a voxel observation of the environment; see Shridhar et al. [7] for more details. The model is trained on 100 demonstrations for each of 18 distinct RLBench tasks [15], annotated with language descriptions with different variations of each.

We use the pre-trained model from [7] to evaluate our approach from Section 2. To reduce computational complexity, we directly use the model output for desired rotation, gripper openness condition, and collision avoidance activation, and we use the uncertainty aware action selection only to modify the desired translation. Further implementation details for Algorithm 1 may be found in Appendix A.

Figure 1 illustrates the value of temperature scaling for improving model calibration on a calibration set of 25 expert demonstrations from the task of interest. The figure shows the reliability diagram for the model [12]. The $y$-axis is the model accuracy at each confidence level, which is the number of predicted actions that are the same as the expert demonstration, divided by the total number of predictions made with confidence within the thresholding interval. The $x$-axis depicts the model confidence, which is maximum value of the probabilty distribution representing the calibrated model: $\max_{a \in \mathcal{A}} \tilde{f}_a(o, c)$. The confidence is perfectly-calibrated when the accuracy and the confidence are linearly related with the slope of 1. The figure depicts this reliability diagram for two tasks, with both the uncailbrated model and the model calibrated using temperature scaling.

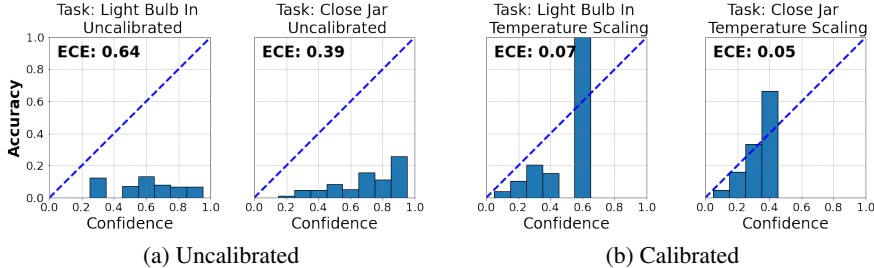

(a) Uncalibrated            (b) Calibrated

Figure 1: Reliability diagrams are shown for two of the tasks. Figure 1a illustrates the reliability diagrams before calibration, while Figure 1b shows the reliability diagrams after running temperature scaling on a calibration set of 25 expert demonstrations from the each task. Overlayed on the plot is the value for the expected calibration error (ECE, See [12] for further details). We see that temperature scaling provides a marked improvement in the calibration.

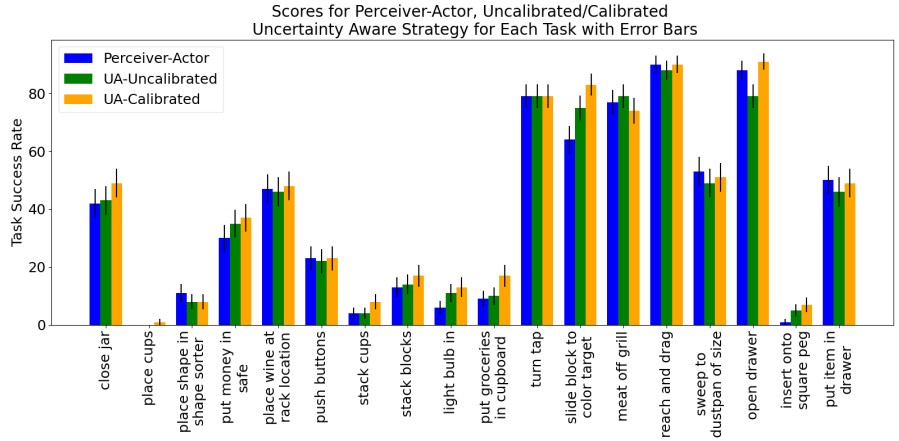

Figure 2: Task Success Rate Score for Original Perceiver-Actor, Uncertainty Aware Strategy with Calibrated/Uncalibrated Confidence for Each Task

Figure 2 shows the success rate of applying Algorithm 1 on each of the 18 tasks that the model was trained on. For each task, we plot the success rate along with bars denoting the standard error for 1) the unmodified output of the Perceiver-Actor model, 2) Algorithm 1 applied to the uncalibrated model, and 3) Algorithm 1 applied to the model calibrated with temperature scaling. The overall success rates for these three approaches are $38.17\%, 38.5\%, 41.39\%$, respectively, each with a standard error of $1.15\%, 1.15\%, 1.16\%$. From this result, we see that compared to greedily selecting the action with the highest probability score in Perceiver-Actor, the uncalibrated uncertainty aware action selection strategy does not demonstrate a significant improvement. However, when applied to the model calibrated with temperature scaling, our method has a $3.22\%$ boost in overall performance, which shows the importance of calibration for uncertainty-aware deployment.

While the modified action is beneficial for most of the tasks, the benefit is most pronounced on tasks with a lower success rate. We hypothesize that this benefit is due to the larger discrepancy between the action with the highest confidence score and the action with the largest sum of neighboring confidence scores on tasks for which the model is overall less confident. Also note that when our method underperforms the baseline, it always falls within one standard error.

## 4   Conclusion

We proposed the use of simple classification model calibration techniques in conjunction with an uncertainty aware action selection approach to improve the success rates for task conditioned imitation learning in robotic manipulation. Our results suggest that appropriately calibrated models lead to improved decision making in this setting. Future work will expand upon the approach to improve transfer to zero-shot tasks and determine whether the approach is effective for other models.

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

# A   Implementation Details

The action space $\mathcal{A}$ for robotic manipulation problems is often very large, and it is computationally expensive to search over the entire action space to calculate the neighboring confidence for each action. Meanwhile, while the probability output from a well-trained model can span the entire action space, there are relatively few entries with high confidence scores. For these reasons, we propose restricting our search to actions that have relatively high confidence scores.

Our method assumes that we have a calibrated model $\tilde{f}$. First we apply a threshold to filter out the actions with probabilities higher than this threshold. Then, if the resulting action space has more than k elements, we keep only the top k of them. After this, we select the center of the remaining actions, define a search space $\mathcal{H}$ around the center with radius $D/2$, and go through each action in this search space. For each action $a$ we get in the search space, we extract the remaining actions $a' \in \mathcal{A}'$ around it with $\|a - a'\|^2 < \tau$, and calculate the accumulative scores for $a$. Finally, we take the $\hat{a}$ that has the greatest accumulative score to be our desired action. This design mitigates the computational problem described above and overlooks the non-informative parts in the model's output, which is formulated in Algorithm 2. In our experiment, we select the robustness threshold $\tau$ and search size $D$ by performing a parameter sweep using cross validation. We choose the probability threshold $\alpha = 1/|\mathcal{A}|$ and $k = 4000$.

---

**Algorithm 2** Detailed Uncertainty Aware Action Selection Strategy

---

**Input:**   Calibrated model $\tilde{f}$, conditioning $c$, observation $o$, robustness threshold $\tau$, probability threshold $\alpha$, search size $D$.
Compute expert action probabilities: $p_1, \ldots, p_{|\mathcal{A}|} = \tilde{f}(o; c)$.
$\mathcal{A}' = \{a \in \mathcal{A}, \ p_a > \alpha\}$,
if $|\mathcal{A}'| > k$:
    keep $k$ actions in $\mathcal{A}'$ with the highest scores
c = mean($a' \in \mathcal{A}'$)
$\mathcal{H} = \{ a \in [c - D/2, c + D/2]^3 \cap \mathcal{A} \}$
Set $\hat{a} = \mathrm{argmax}_{a \in \mathcal{H}} \sum_{a' \in \mathcal{A}'} p_{j(a')} \mathbf{1}(\|a - a'\|^2 < \tau)$
**Output:**   Uncertainty-aware action $\hat{a}$

---

