# OpenReview forum: "Uncertainty Aware Deployment of Pre-trained Task Conditioned Imitation Learning Policies"
_robot-learning.org/CoRL/2023/Workshop/OOD — OOD Workshop @ CoRL 2023_

### Official Review · Reviewer_cmcZ · 2023-10-13
**Interesting work with potential**

**Rating:** 7
**Confidence:** 5

**Review:**

The paper proposes calibrating the imitation-learned output with temperature scaling, and the final output is chosen as the action with highest nearby robustness, which is more robust to errors and requires good calibration of the model. The technique offers small improvement in the 18 RLBench tasks.

The story of the work is clear and well-motivated. The technique is novel while the choice of the action taking over the nearby region is justified, in my opinion. Although the task performance improvement is small, I imagine it will be larger if the authors consider more challenging settings, especially real environments with noisy observations.

It will also be interesting to consider a multi-step setting where the model output is the expected future value. I imagine the error from miscalibration compounds, and the importance of calibration is even more significant.

---

### Official Review · Reviewer_rJss · 2023-10-16
**Performance improvement of IL policies using uncertainty quantification**

**Rating:** 7
**Confidence:** 4

**Review:**

This work proposes a method for calibration of pretrained models using temperature scaling and an uncertainty aware action selection algorithm to improve out-of-distribution generalization for imitation learning tasks. The work shows a ~3% improvement over uncalibrated models. The paper is well-written and the results are promising.

I would have liked to see an added discussion on why the calibrated model performs worse in some scenarios as seen in Figure 2. It seems to me that the calibration step should only help improve performance. Why is the UA-Calibrated performance worse than the UA-uncalibrated performance and that of the original Perceiver-Actor on tasks that already have lower discrepancy?

Given that the tasks considered are what the model was trained on, it's unclear how generalization is achieved in this work - which is what the authors claim to address in the abstract. A discussion on what the authors mean by generalization in this context would be helpful.

The authors also discuss conformal prediction techniques for calibration in the related work. Could a CP-based calibration be used here? If so, a comparison with CP-based calibration could be a potential avenue of future work to assess the efficacy of this approach.

---

### Decision · Program_Chairs · 2023-10-17

**Decision:**

Accept

**Comment:**

We agree with the reviewers’ assessment that this work is technically sound and will contribute to productive, topical discussions at the 2023 Workshop on OOD Generalization in Robotics. We appreciate that your work highlights the role of calibration in achieving good generalization performance at deployment time. We recommend the authors incorporate the reviewers’ feedback into their camera-ready submission to further improve their manuscript.